# A Combination of the Immunotherapeutic Drug Anti-Programmed Death 1 with Lenalidomide Enhances Specific T Cell Immune Responses against Acute Myeloid Leukemia Cells

**DOI:** 10.3390/ijms24119285

**Published:** 2023-05-26

**Authors:** Barbara-ann Guinn, Patrick J. Schuler, Hubert Schrezenmeier, Susanne Hofmann, Johanna Weiss, Christiane Bulach, Marlies Götz, Jochen Greiner

**Affiliations:** 1Centre for Biomedicine, Hull York Medical School, University of Hull, Hull HU6 7RX, UK; barbara.guinn@hyms.ac.uk; 2Department of Otorhinolaryngology, University Hospital Ulm, 89081 Ulm, Germany; patrick.schuler@uniklinik-ulm.de; 3Institute of Transfusion Medicine, University of Ulm and German Red Cross, 89073 Ulm, Germany; h.schrezenmeier@blutspende.de; 4Department of Internal Medicine V, University Hospital Heidelberg, 69120 Heidelberg, Germany; susanne.hofmann@med.uni-heidelberg.de; 5Department of Internal Medicine III, University Hospital Ulm, 89081 Ulm, Germany; johanna.weiss95@gmail.com (J.W.); christiane.bulach@outlook.com (C.B.);; 6Department of Internal Medicine, Diakonie Hospital Stuttgart, 70176 Stuttgart, Germany

**Keywords:** acute myeloid leukemia, anti-programmed death 1 (PD-1), lenalidomide, immunotherapy, leukemia-associated antigens

## Abstract

Immune checkpoint inhibitors can block inhibitory molecules on the surface of T cells, switching them from an exhausted to an active state. One of these inhibitory immune checkpoints, programmed cell death protein 1 (PD-1) is expressed on T cell subpopulations in acute myeloid leukemia (AML). PD-1 expression has been shown to increase with AML progression following allo-haematopoeitic stem cell transplantation, and therapy with hypomethylating agents. We have previously shown that anti-PD-1 can enhance the response of leukemia-associated antigen (LAA)-specific T cells against AML cells as well as leukemic stem and leukemic progenitor cells (LSC/LPCs) ex vivo. In concurrence, blocking of PD-1 with antibodies such as nivolumab has been shown to enhance response rates post-chemotherapy and stem cell transplant. The immune modulating drug lenalidomide has been shown to promote anti-tumour immunity including anti-inflammatory, anti-proliferative, pro-apoptotic and anti-angiogenicity. The effects of lenalidomide are distinct from chemotherapy, hypomethylating agents or kinase inhibitors, making lenalidomide an attractive agent for use in AML and in combination with existing active agents. To determine whether anti-PD-1 (nivolumab) and lenalidomide alone or in combination could enhance LAA-specific T cell immune responses, we used colony-forming immune and ELISpot assays. Combinations of immunotherapeutic approaches are believed to increase antigen-specific immune responses against leukemic cells including LPC/LSCs. In this study we used a combination of LAA-peptides with the immune checkpoint inhibitor anti-PD-1 and lenalidomide to enhance the killing of LSC/LPCs ex vivo. Our data offer a novel insight into how we could improve AML patient responses to treatment in future clinical studies.

## 1. Introduction

Acute myeloid leukemia (AML) is defined as a malignant disorder of the bone marrow characterised by the clonal expansion and differentiation arrest of myeloid progenitor cells. Chemotherapy is the standard treatment for AML but alone cannot stave off the high frequency of relapse seen in adults [1]. Some of the immunotherapy treatments, now routinely used to treat AML patients, such as allogeneic hematopoietic stem cell transplantation and donor lymphocyte infusion, offer significant success and risk, whereas other immunotherapeutic approaches needed to enhance outcomes have only recently entered clinical practice and need to be further developed (reviewed in detail elsewhere [2]).

Single antibody treatments have been considered to be one of the most promising treatment options for cancers whose tumour antigen is surface expressed. However, antibody therapies, such as anti-CD33 or anti-CD38, have been shown to have only modest effects in clinical trials for AML (reviewed in [3]). While unconjugated antibodies can stimulate NK cells, antibody-dependent cell-mediated cytotoxicity (ADCC) bi- and tri-specific antibodies can engage either NK or T cells to redirect anti-AML immune responses in a highly specific manner. Potency can be further increased through the use of conjugates such as toxins or radio-chemicals. However, relapse remains the major issue, caused by clonal evolution [4] and the immunosuppressive effects of the bone marrow microenvironment [5], both of which contribute to subsequent immune escape.

Immune checkpoint inhibitors (ICIs) act as positive modulators of the immune system, and are being used as induction therapy to enhance response rates and to improve relapse-free survival post-chemotherapy and bone marrow transplantation (recently reviewed by [6]). Programmed cell death protein 1 (PD-1) and CTLA-4 (also known as CD152) are the most widely investigated checkpoint receptors [7]. Single anti-PD-1 monoclonal antibody infusions show only modest clinical efficacy [8]; however, in combination with hypomethylating agents, they represent promising treatments for relapsed/refractory AML patients as well as elderly patients as a first-line therapy option (recently reviewed in [9]). Treatment works by blocking PD-1/PD-L1 signaling between antigen presenting cells (APCs) and T cells (Figure 1). In doing so, they allow effective T cell responses, specific to peptide presented in the context of MHC class I, to occur.

Lenalidomide is a synthetic compound derived from structural modifications of thalidomide, itself derived from glutamic acid. Thalidomide was banned in the 1960s because of reports of congenital abnormalities in babies whose mothers had taken it in trimester 1 to manage morning sickness. It is believed to cause teratogenicity through oxidative stress with subsequent downregulation of the Wnt and Akt survival pathways causing apoptosis during early fetal limb development [11]. However, three decades later thalidomide was shown to improve erythema nodosum leprosum, an immune complex mediated inflammatory reaction that occurs in leprosy patients, during treatment. Research into the mechanism of action of thalidomide showed its immunological and immunomodulatory effects; however, because of adverse effects such as neuropathy, deep vein thrombosis and sedation, safer analogues were developed. Lenalidomide, unlike thalidomide, is not teratogenic in rabbit models, has a lower incidence of adverse effects and is 100–1000 times more potent at stimulating T cell proliferation as well as IFN-γ and IL-2 production. It has been used to treat adults with multiple myeloma, smoldering leukemia, low risk myelodysplasia with a 5q deletion and AML [12,13,14,15].

Lenalidomide has multiple activities that promote anti-tumour immunity including the ability to inhibit tumour necrosis factor (TNF), for which it was first identified, an anti-inflammatory effect, anti-proliferative and pro-apoptotic effects seen in multiple myeloma cells in vitro and anti-angiogenic activity [16]. It also induces activation of cell adhesion molecules, T cell proliferation, IL-2 and interferon-γ production, and enhances the cytotoxic activity of natural killer cells [17]. These effects are each distinct from chemotherapy, which is based on nucleoside analogues (such as cytarabine or fludarabine) and anthracyclines (such as daunorubicin and idarubicin), hypomethylating agents (such as decitabine and azacitidine) or kinase inhibitors, making lenalidomide an attractive agent for use in AML in combination with existing active agents. However, its efficacy in AML remains inconclusive [14]. A phase III study in which lenalidomide either was or was not integrated into standard induction therapy for newly diagnosed adult AML patients (aged 18-65 years) showed no difference in outcomes for patients at a median follow-up period of 41 months for event-free survival or overall survival [18]. However, there was some evidence that lenalidomide may benefit SRSF2-mutant AML patients.

In immunotherapy, it is feasible to use combinations of different therapeutic options [19] mostly because their targets differ. This is especially the case for the ICI, anti-PD-1, and the immune modulatory agent (IMiD), lenalidomide [20,21]. Lenalidomide treatment has been used with the expectation of modulating immune responses in AML with mixed results [14,18,22]. In older previously untreated AML patients, treatment with azacytidine and lenalidomide led to an overall response rate of 40% while early death was noted in 17% of patients [23]. However, the impact of an IMiD and ICI to attenuate LAA-specific immune responses has not been studied until now. We have previously shown that anti-PD1 enhances the cytotoxic effect of leukemia-associated antigen (LAA)-stimulated cytotoxic T-lymphocytes (CTLs) against leukemic progenitor and stem cells (LPC/LSCs) and this was most noticeable against NPM1^mut^ AML cells when the immunogenic epitope was derived from the mutated region of NPM1 [24]. In this study, we wanted to investigate whether lenalidomide could further increase LPC/LSC destruction of NPM1^mut^ AML cells in the presence of anti-PD-1.

## 2. Results

### 2.1. Immunoassays

We used functional CFI to investigate the effect of the checkpoint-inhibitor anti-PD-1 (nivolumab) or lenalidomide, alone or in combination on the antigen-specific immune responses via LAA-stimulated specific T cells against leukemic cells and also LPC/LSC taken from AML patients.

An inhibition in colony-forming units was apparent following the addition of anti-PD-1 to CTL for several days before starting CFI. CTL from HDs led to an inhibition in CFI when adding anti-PD-1 to nine of the 16 patient samples analysed. Lenalidomide led to a decrease in colonies in eight of 16 patients and the combination of both anti-PD-1 and lenalidomide augmented this effect with 11 out of 16 patients showing a reduction in colony numbers (Figure 2A). We could see a difference in AML NPM1^WT^ patients (Figure 2B) compared to NPM^mut^ patients (Figure 2C).

There was a decrease in the number of colonies that grew in the presence of NPM1 or PRAME P300 and lenalidomide, regardless of whether the cells were derived from NPM^mut^ and NPM^WT^ patients (Figure 3). To extend our understanding of the impact of peptide stimulation on the immune killing of patient cells, we incubated patient samples with LAA-peptide and showed that this alone was sufficient to cause a reduction in colony numbers. The addition of anti-PD-1 and/or lenalidomide augmented the reduction of colonies when patient samples were treated with LAA specific for NPM1^mut^ or NPM1^WT^.

There was a decrease in the number of colonies that grew in the presence of PRAME, WT-1 or NPM1 and lenalidomide, impacting cells derived from NPM^WT^ and NPM^mut^ patients (Figure 3). For a better understanding, we have only shown the CFI results from patient cells where there was a reduction in colony numbers following LAA-peptide stimulation alone.

### 2.2. ELISpot Assays

Sixteen AML patients were tested who showed an immune response against at least one of these antigens (PRAME, WT-1 or NPM-1; Pt 1-4 PRAME, Pt 5-8 WT-1 and Pt 9-16 NPM1). The samples were then stimulated with the LAA that they showed the strongest response against [24] and we tested lenalidomide alone or in combination with the anti-PD-1 antibody in IFNγ ELISpots. All data were normalised to the respective LAA. For the NPM1^WT^ patients, the addition of anti-PD-1 to patient 1 resulted in a 2-fold higher response, lenalidomide in a 4.6-fold and the combination of both in a 4.2-fold increased response. Patient 8 showed responses to anti-PD-1 (1.8-fold), Lena (1.3-fold) and in combination (1.3-fold). For the NPM1^mut^ patients, Pt 1/2/4 showed some responses, anti-PD-1 1/1.0/1.0-fold, Lena 1.1/1.0/1.8-fold and in combination 1.0/1.8/1.0-fold, respectively. The data are not comparable with the results of CFI assays, but we could see stimulation due to the addition of the drugs in the ELISpots (Figure 4). We tested immune responses against leukemic cells, focusing on LPC/LSC (Figure 2 and Figure 3).

Interestingly, in the wildtype patients who responded, the synergistic effect of PD-1 in combination with lenalidomide cannot be seen clearly (Figure 2D). However, a synergistic effect was clearer in the mutated patients, where the average colony reduction was 40% for anti-PD-1, 52% for lenalidomide and 61% for the combination of anti-PD-1 and Lena (Figure 2E). In the ELISpot assays, some of the effects observed in CFI could be replicated; e.g., in patient 1, who was NPM^WT^, a response to anti-PD-1/Lena and the combination of both was observed and patient 7 responded to PD-1/Lena and the combination. Patient 2, who had a NPM1^mut^, showed a small response to PD-1 and Lena alone in the CFI, but the combination exhibited a strong response.

## 3. Discussion

AML is a heterogenous disease with many subtypes and often with complex underlying genetic abnormalities [25,26]. Treatment options have improved in recent years and complete remission rates are close to 80%; however, 50% of patients will relapse due to the escape of drug-resistant clones [4] with poor associated survival. Since 2017, there has been a rapid growth in the treatment options available to patients with approval of a number of drugs, including kinase and isocitrate dehydrogenase (IDH) inhibitors, anti-CD33 drug conjugates and the cytotoxic chemotherapy CPX-351 (reviewed in [27]). Papeammanuil et al. [28] and Prieto-Conde et al. [29] have evidenced the capacity of genetic analysis (qPCR and the more discovery-powerful Next Generation Sequencing (NGS [30])) to risk stratify AML patients and inform clinical decisions. However, further improvements in treatments are still necessary, especially for older people who suffer from AML, since they often have a dismal prognosis [31] and many new treatments are too severe for this patient group [32].

Immunotherapy in the form of bone marrow transplants has been used to treat leukemia patients for almost 70 years. Immunotherapy has now evolved to include peripheral blood stem cell transplants, donor leukocyte infusions, monoclonal antibodies, adoptive T and/or NK cell therapy, checkpoint blockade and leukemia vaccines (reviewed in [33]). Increasingly, conventional and immunotherapy strategies are being used in combination, often to treat relapse and/or patients who have become refractory to existing treatments [34]. Recently, Short et al. [35] enrolled 50 patients into one of six different combinations of the hypomethylating agent azacitidine and antibodies that target CD33, OX40, PD-L1, BCL-2 or the Hedgehog pathway. In addition to showing the most promising combination of treatments for patients who had relapsed/refractory AML, the multi-arm trial was able to efficiently evaluate novel therapies. Current open clinical trials include CCS1477 that blocks the action of p300 and cyclic-AMP response element binding protein, and venetoclax and low dose cytarabine with intensive chemotherapy for NPM^mut^ AML patients over 60 years of age.

One of the first adoptive therapy clinical trials was performed on an AML patient (allogeneic transplantation) and led to a high rate of durable remissions in those who receive them. This demonstrates that leukemia could be eradicated by an effective immune response and the effectiveness of the immune response understandably plays an increasingly prominent role in treatment decisions in AML [36,37]. However, the role of checkpoint inhibitors, mainly those targeting T cells such as nivolumab and pembrolizumab or those targeting macrophages such as the anti-CD47 antibody magrolimab, are still under investigation. Checkpoint inhibitor monotherapies have shown little clinical activity in high-risk AML [38], while early-phase clinical trials suggest that ICIs and hypomethylating agents are safe and more promising [37], again with the exception of high-risk AML patients [39].

Some concerns about side effects from the use of lenalidomide have been raised. Lenalidomide is an immunomodulatory agent (IMiDs) and as such it binds to cereblon and activates cereblon E3 ligase activity, resulting in the rapid ubiquitination and degradation of two specific B cell transcription factors, Ikaros family zinc finger proteins Ikaros (IKZF 1) and Aiolos (IKZF3) [40]. These may cause direct cytotoxicity by inducing free radical mediated DNA damage. IMiDs also have anti-angiogenic, immunomodulatory and tumour necrosis factor alpha inhibitory properties [41]. Lenalidomide causes neutropenia and thrombocytopenia in some patients [42]; however, these side effects are often mild (grade I and II) and clinically manageable. From the ELISpots in this study, we can see the functionality of the T cells is preserved, and in the positive control there was no decrease in cell number plus/minus immunotherapeutics, even when the treatments were combined. The clinical side effects of the ICI anti-PD-1 (nivolumab) are immunotoxicities such as pneumonitis, nephritis, immune-related rash and transaminitis. Side effects in the bone marrow/hematopoietic system can be thrombocytopenia, haemolytic anaemia, neutropenia, aplastic anaemia, pure red cell aplasia and hemophagocytic lymph histiocytosis. As ICI indications for patients with melanoma and other tumour types continue to broaden, more patients will receive ICI therapies. Therefore, the number of clinically significant toxicities will undoubtedly increase, although the incidence of severe immune-related adverse events has been considered relatively low still, especially for the ICI Nivolumab [43].

Pathogenic variants in NPM1 are among the most common in AML. AML patients with NPM^mut^ were recognised as a distinct entity by the World Health Organisation in 2017, and the disease was associated with a favourable prognosis in the absence of FLT3 ITD [44]. NPM1^mut^ can be targeted directly or through the pathways it interacts with (reviewed in [45]) using drugs such as Venetoclax, a selective Bcl-2 inhibitor which when used in combination with 5′-azacytidine has shown anti-leukemic activity in 60–70% of AML patients [46]. Indeed, venetoclax and 5′aza-cytidine, or decitabine and venetoclax, and low dose cytarabine have become the standard treatment for previously untreated older and unfit AML patients [46].

## 4. Materials and Methods

### 4.1. Sample Preparation, Isolation and Freezing

Peripheral blood samples from eight AML NPM1^WT^ and eight AML NPM1^mut^ were further evaluated with anti-PD-1 (nivolumab) in combination with the immunomodulating drug lenalidomide. Patient samples were taken following informed consent and in accordance with the Declaration of Helsinki. The local ethics committee (No. 334/09 and No. 221/14) approved the study protocol. Peripheral blood mononuclear cells (PBMCs) from healthy donors (HDs) were separated via Ficoll (Pan Biotech, Aidenbach, Germany) density gradient centrifugation, cryopreserved and stored in liquid nitrogen. All patient samples consisted of more than 90% leukemic blasts. Healthy volunteer samples were obtained from the German Red Cross Ulm.

### 4.2. Viral- and Leukemia-Associated Antigens

The following LAAs were chosen according to previous analyses and positivity in colony-forming immunoassays (CFIs) [47], PRAME (P300 (ALYVDSLFFL)), WT1 (RMFPNAPYL) and NPM1 (AIQDLCVAV) for AML NPM1^mut^ patients only. All peptides were HLA-A2-restricted.

### 4.3. Patients’ Characteristics and Selection of LAAs

To define the potential of specific CTLs, allogeneic T cells were stimulated first with various LAAs (Section 4.2), then the LAA with the strongest response was chosen for further rounds of stimulation. Only HLA-A2 positive patient samples and HDs were used, since all LAAs were HLA-A2-restricted. We then added the checkpoint inhibitor anti-PD-1 or lenalidomide alone, or the combination of both.

### 4.4. Mixed Lymphocyte Peptide Cultures (MLPCs)

In MLPCs, peptide-specific CD8+ allogeneic T cells were generated from HD samples, providing effector cells (E) for further assays. Briefly, samples were thawed, counted and divided into two portions. One fraction, employed as APCs, were irradiated with 30 Gy and pulsed with the respective peptides for 1.5 h at 37 °C. Thereafter, APCs were mixed with the second fraction E at a ratio of 1:1. On the second day, IL-2 (2.5 ng/mL) and IL-7 (20 ng/mL) were added, and the appropriate drugs were added to some wells and incubated for seven to nine days and then used for functional assays.

### 4.5. Addition of Lenalidomide to Cell Culture

In line with the results of a former titration [48], 5 µg of the anti-PD-1 antibody (nivolumab) and/or 5 µg of lenalidomide, which corresponds to the median dose that patients receive (10 mg lenalidomide per day), was added to the MLPC on day 0 to the respective wells containing the E fraction for one hour then the irradiated and stimulated APC fraction was added. In this way, the direct effects of the ICI anti-PD-1 and/or the IMiD lenalidomide on CD8+ T cells were measured. CD8+ T cells were stimulated with CMV or the respective LAA only as a control.

### 4.6. Colony-Forming Immunoassays

Allogeneic T cells from MLPC were used as effectors and the ratio of E:Target (T) was 10:1. Primary patient T cells were used as a source of E and T cells stimulated with no peptide served as a growth control. E and T were incubated together at 37 °C for 4 h, resuspended in IMDM-Medium containing 2% FCS and added to a 3 mL HSC-colony-forming unit complete medium (Miltenyi Biotech, Bergisch Gladbach, Germany), then aspirated using a syringe. A total of 1.1 mL medium was placed into each cell culture dish (Thermo Scientific, Waltham, MA, USA). Colonies were analysed after a 20-day incubation time; the difference between control and sample in percent was calculated and displayed.

### 4.7. ELISpot (Enzyme-Linked-Immuno-Spot)

Membrane bottom 96 well plates were coated with a solid antibody phase. Subsequently, the membranes were incubated with allogeneic pre-stimulated peripheral blood lymphocytes from MLPC and APC from leukemia patients at a ratio of 5:1. Cytokines bound to the solid antibody phase were visualised with specific antibodies coupled to biotin, alkaline phosphatase and the corresponding substrate. The evaluation was carried out using an ELISpot reader. IFNγ (Mabtech, Nacka Strand, Sweden) ELISpot was performed according to the manufacturer’s instructions.

### 4.8. Statistical Analysis

Statistical tests were performed using GraphPad PRISM v8. The program was also used to evaluate assays, for comprehensive analysis, for organising data and for graphing. As a statistical analysis, we used an ordinary one-way ANOVA test and used * = *p* < 0.05, ** = *p* < 0.01, *** = *p* < 0.001 and **** = *p* < 0.0001.

## 5. Conclusions

We have previously shown that anti-PD-1 antibody alone can increase LAA stimulated cytotoxic T lymphocytic responses and their cytotoxic effect against LPC/LSC [48]. In this study, we examined anti-PD-1 (nivolumab) together with lenalidomide in vitro, using primary cells from AML patients. We could see a response for anti-PD-1 in all patient groups and improved results when a combination of the ICI anti-PD-1 and the IMiD lenalidomide were used. Results using the combination showed enhanced LPC/LSC killing particularly in the NPM1^mut^ AML patient group. The effect was strongest against NPM1^mut^ cells when the immunogenic epitope was derived from the mutated region of NPM1 and these effects were enhanced through the addition of anti-PD-1 [24]. We were able to show that in combination these two agents could act together to further kill colonies derived from NPM^WT^ and NPM^mut^ AML patients.

Taken together, combinations of immunotherapeutic approaches increase antigen-specific immune responses against leukemic cells but also LPC/LSC, especially the combination of LAA-peptides with the anti-PD-1 antibody and one further immunomodulating drug, providing an interesting option for further clinical studies.

## Figures and Tables

**Figure 1 ijms-24-09285-f001:**
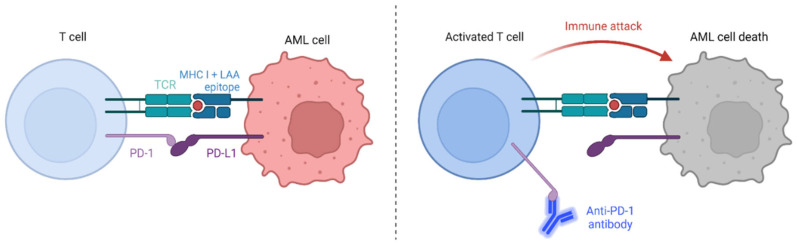
The immune checkpoint, PD-1:PD-L1 interaction, inhibits T cell activation. However, the use of anti-PD-1 antibodies such as nivolumab act as ICIs that allow tumour cell destruction to proceed. We propose that the addition of the immunomodulatory drug lenalidomide will further enhance anti-leukemia T cell responses, leading to more effective LSC/LPC death. Adapted from “Immune checkpoint inhibitor against tumor cell” by BioRender (2023) [10]. Retrieved from https://app.biorender.com/biorender-templates.

**Figure 2 ijms-24-09285-f002:**
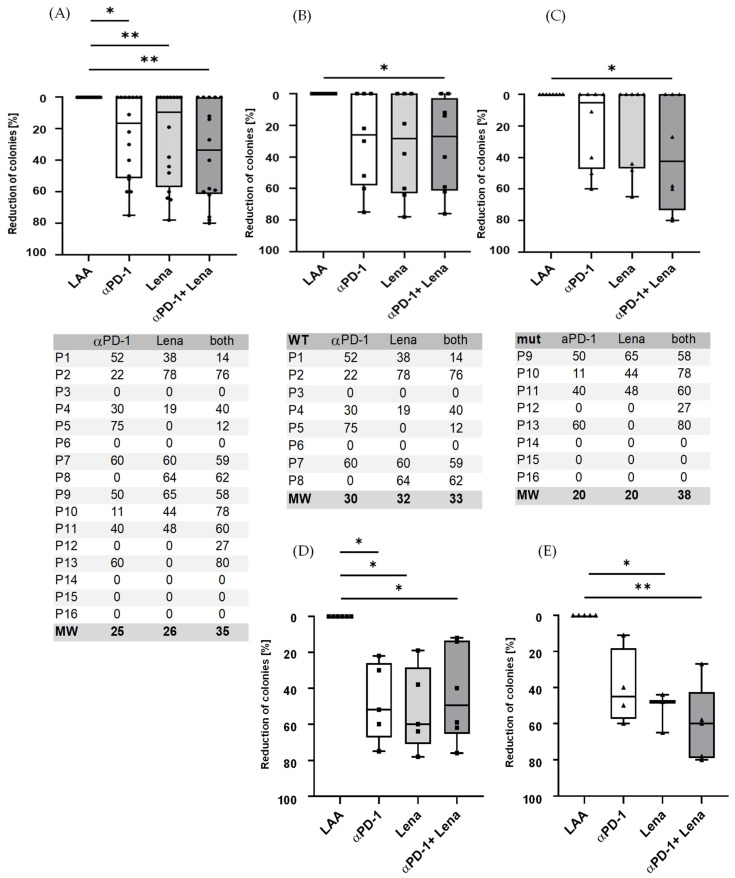
Immune responses increased following the addition of anti-PD-1 (nivolumab) to stimulated LAA-specific T cells. (**A**) Nine samples from the 16 patient samples analysed showed a significant reduction in the number of colonies when treated with anti-PD-1 with an average reduction of 25% (range 0–75%), for Lena alone 8/16 patients with average reduction of 26% (range 0–78%) and for the combination of anti-PD-1 and Lena the reduction in 11/16 patients was 35% (range 0–80%). (**B**) All eight NPM1^WT^ patient samples showed a reduction for anti-PD-1/Lena/both of 20/20/38% and (**C**) all eight NPM1^mut^ patients for anti-PD-1/Lena/both of 30/32/33%. (**D**,**E**) Responders only. (**D**) Respectively, 5/5/6 patients of eight NPM1^WT^ patients were responders when treated with anti-PD-1/Lena/both in combination; the average reduction rate was 48/52/44%. (**E**) Of eight NPM1^mut^ patients, 4/3/6 patients were responders for PD-1/Lena/PD-1 and Lena in combination. The mean reduction was 40% in NPM1^mut^ patients for anti-PD-1 (range 40–60% reduction). Lenalidomide had an effect on 52% (range 44–65% reduction) in NPM1^mut^ patients. The combination of lenalidomide and anti-PD-1 in responders was 61% (range 27–80%) in NPM1^mut^. The inhibition in CFI was calculated for each patient and antigen separately; thus, the anti-PD-1 effect was shown to be patient-/antigen-dependent. * = *p* < 0.05, ** = *p* < 0.01. The tables below (**A**–**C**) show the reduction in the number colonies as a percentage (%) for each figure above.

**Figure 3 ijms-24-09285-f003:**
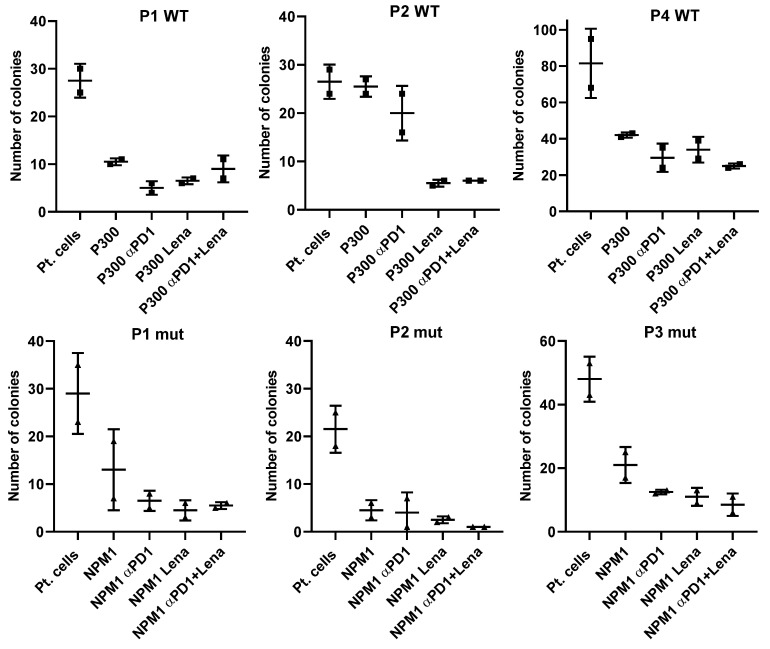
Three exemplar NPM^WT^ and three NPM^mut^ patients show the effect of anti-PD1 (αPD1), lenalidomide (Lena) and the combination of both on colony formation in the presence of LAA stimulation. To understand the whole design of the tests better, we have shown the colony-formation capability of each patient without any LAA or treatment (pt. cells), and also the reduction in colonies in the presence of the LAA (PRAME) that stimulated T cells in these patients the most, and then with addition of the drugs alone or in combination.

**Figure 4 ijms-24-09285-f004:**
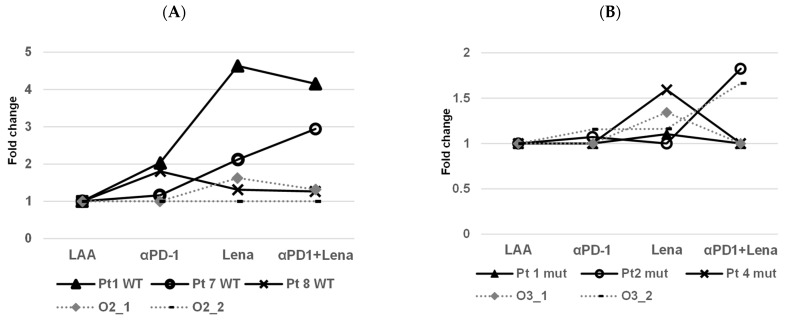
Examination of IFNγ production via CD8+ T cells from three NPM1^WT^ patients (Pt) and a WT cell-line (Oci-AML-2) and three NPM1^mut^ patients and an NPM1^mut^ cell-line (Oci-AML-3). In all ELISpot results, the fold change was calculated. (**A**) Patient 1 was NPM1^WT^ (Pt1 WT) and showed augmented stimulation with anti-PD-1 (2-fold), lenalidomide (Lena 4.6-fold) and the combination of both (4.2-fold). Patient 7 with NPM1^WT^ (Pt 7 WT) showed a low stimulation with anti-PD-1 (1.2-fold), with Lena (2.1-fold) and with the combination of both (2.9-fold). Patient 8 with NPM1^WT^ (Pt 8 WT) showed augmented stimulation with anti-PD-1 (1.8-fold), Lena and the combination of both (both at 1.3-fold). In one assay, Oci-AML-2 showed a 1.6-fold response with Lena and 1.3-fold with the combination of both. (**B**) Patient 1 NPM1^mut^ (Pt 1 mut) showed slightly augmented stimulation with Lena alone (1.1-fold) and/or the combination of both drugs, anti-PD-1 exhibited no stimulation. Pt 2 NPM1^mut^ (Pt 2 mut) showed a low stimulation with anti-PD-1 (1.1-fold), with the combination of both (1.8-fold) but not with Lena alone (1.0-fold). Patient 4 NPM1^mut^ (Pt 4 mut) showed an effect with Lena only (1.6-fold). Oci-AML-3 showed a response for Lena only (1.3-fold) in the first assay. In the second assay, anti-PD-1 and Lena showed an effect (each 1.2-fold) and with the combination of both a stimulation of 1.7-fold was seen. In all ELISpot results, the fold change was calculated taking as a basis the respective LAA alone.

## Data Availability

The data that support the findings of this study are available from the corresponding author or from marlies.goetz@alumni.uni-ulm.de upon reasonable request.

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
