# Peer review of "A Combination of the Immunotherapeutic Drug Anti-Programmed Death 1 with Lenalidomide Enhances Specific T Cell Immune Responses against Acute Myeloid Leukemia Cells"

_ijms, 2023, doi:10.3390/ijms24119285_

Round 1

Reviewer 1 Report

In this manuscript, the authors find that anti-PD-1 and lenalidominde combination treatment can enhance the T cell immune response against AML cells using patient samples. It's an interesting story. However, they only did colony assay and IFNgamma ELISpot to show the responses. More T cell immune response analysis is recommended. It will be better to know if anti-PD-1 and lenalidominde has synergistic effect. And the side effect also need to be considered.

Author Response

Reviewer 1:

In this manuscript, the authors find that anti-PD-1 and lenalidominde combination treatment can enhance the T cell immune response against AML cells using patient samples. It's an interesting story. However, they only did colony assay and IFNgamma ELISpot to show the responses. More T cell immune response analysis is recommended.

We thank the reviewer for this important comment. We have performed more ELISpots. Due to limited patient cells we could now include in total 3 NPM1 WT and 3 NPM1 mut patients and also the NPM1 WT cell-line Oci-AML-2 and the NPM1 mut cell-line Oci-AML-3. Please see text on page 8 and new Figure 4 A and B.

It will be better to know if anti-PD-1 and lenalidominde has synergistic effect.

We thank the reviewer. We have included Figure 2D and E to demonstrate what we have stated in the text regarding the responders and the effects of anti-PD-1 and Lenalidomide alone and the synergistic effects. Naturally, not all patients respond to a drug, in our assays it is the same. In Figure 2E we are able to show that there are indeed synergistic effects with anti-PD-1 and lenalidominde in combination, but understandably only in responders and the effect is most notable in NPM1 mut patients. Please see the new figures 2D and E.

We have added two further articles in the Introduction (Line 120) that show, that Lenalidomide can enhance the immune effect of anti-PD-1:

  1. Kwon et al. Two-Round Mixed Lymphocyte Reaction for Evaluation of the Functional Activities of Anti-PD-1 and Immunomodulators. Immune Netw. 2018 Dec 19;18(6):e45.  doi: 10.4110/in.2018.18.e45. eCollection 2018 Dec.
  2. Görgün et al. Lenalidomide Enhances Immune Checkpoint Blockade-Induced Immune Response in Multiple Myeloma. Clin Cancer Res. 2015 Oct 15;21(20):4607-18. doi: 10.1158/1078-0432.CCR-15-0200. Epub 2015 May 15.

And the side effect also need to be considered.

We thank the reviewer for mentioning this point. We have added the following paragraph to the discussion to address the reviewers request:-

Some concerns about side effects from the use of lenalidomide have been raised. Lenalidomide is an immunomodulatory agent (IMiDs) and as such, it binds to cereblon and activates cereblon E3 ligase activity, resulting in the rapid ubiquitination and degradation of two specific B cell transcription factors, Ikaros family zinc finger proteins Ikaros (IKZF 1) and Aiolos (IKZF3)[42]. These may cause direct cytotoxicity by inducing free radical mediated DNA damage. IMiDs also have anti-angiogenic, immunomodulatory and tumor necrosis factor alpha inhibitory properties [43]. Lenalidomide causes neutropenia and thrombocytopenia in some patients [44] however these side effects are often mild (grade I and II) and clinically manageable. From the ELISpots in this study we can see the functionality of the T cells is preserved, and in the positive control there was no decrease in cell number plus/minus immunotherapeutics,  even when the treatments were combined. The clinical side effects of the ICI anti-PD-1 (nivolumab), are immunotoxicities such as pneumonitis, nephritis, immune-related rash and transaminitis. Side effects in the bone marrow/hematopoietic system can be: thrombocytopenia, haemolytic anaemia, neutropenia, aplastic anaemia, pure red cell aplasia, hemophagocytic lymph histiocytosis. As ICI indications for patients with melanoma and other tumor types continue to broaden, more patients will receive ICI therapies. Therefore, the number of clinically significant toxicities will undoubtedly increase, although the incidence of severe immune-related adverse events has been considered relatively low still, especially for the ICI Nivolumab [45].

Reviewer 2 Report

Guinn and colleagues based on their data propose that "A combination of the immunotherapeutic drug anti-programmed death 1 with lenalidomide enhances specific T cell immune responses against acute myeloid leukemia cells".

The concept of combining immunomodulatory drugs to enhance T-cell specific effects in order to achieve longterm remission in AML sounds intriguing. The authors here present an in vitro study based on previously used techniques and established methods by this group. 

The manuscript is largely well written. Some parts though would need further clarification and context.

1. The proposed enhanced killing effect of Lenalidomide and the used anti-PD-1 antibody is hard to see, when looking at the presented data. Although there might be a higher number of responders, the cumulative effect in Figure 2 is not visible. For the entire cohort and the NPM out group, single treatment as well as the combination are equally effective. If, at all, there is a higher effect for the NPM1 WT group (Fig 2, and Fig 3 P2 WT). The direct effect on colon formation shown in figure 3 is also not enhanced by the combination with the aforementioned exception. 

This needs to be clarified extensively, as the presented data might show a higher number of responders, but not a true additive effect and the drawn conclusion is not supported. 

Notably,  Figure 2 is not aligned, the legends in the manuscript look somewhat different and the tables included are not intuitive: what numbers are presented here? The percentage of CSF reduction for each individual patient? Also, Figure 3, first row seems to be missing some parts and the y axis is not readable for patients 2, 4, 10 and 11.

2. In context with the first point, the authors should discuss in more depth, how they expect these results to translate to clinical trials, as there have been basically two larger negative trials including Len in the induction treatment of AML patients (with of which have been quoted). Again, context and clarification are essential, to see novelty in this work. 

3. The ELISpot data needs some more context. The authors present data on three of sixteen tested patients. What happened with the other patients? How do the authors interpret these results?

Minor points

specific which anti-PD1 antibody was used

line 124: is this the same as Ficoll?

line 179: delete one  "colony forming"

lines 192: Sentence starting with "However": I do not understand this sentence; where is this shown?

line 255: stratify

line 294: There are hardly any patients with a normal karyotype if analyzed with modern techniques (WGS, OGM); in my very personal opinion I would not emphasize this. 

Author Response

Reviewer 2:

Guinn and colleagues based on their data propose that "A combination of the immunotherapeutic drug anti-programmed death 1 with lenalidomide enhances specific T cell immune responses against acute myeloid leukemia cells".

The concept of combining immunomodulatory drugs to enhance T-cell specific effects in order to achieve longterm remission in AML sounds intriguing. The authors here present an in vitro study based on previously used techniques and established methods by this group. 

The manuscript is largely well written. Some parts though would need further clarification and context.

We appreciate the positive comments and clear statements of the reviewer.

  1. The proposed enhanced killing effect of Lenalidomide and the used anti-PD-1 antibody is hard to see, when looking at the presented data. Although there might be a higher number of responders, the cumulative effect in Figure 2 is not visible. For the entire cohort and the NPM mut group, single treatment as well as the combination are equally effective. If, at all, there is a higher effect for the NPM1 WT group (Fig 2, and Fig 3 P2 WT). The direct effect on colon formation shown in figure 3 is also not enhanced by the combination with the aforementioned exception. 

This needs to be clarified extensively, as the presented data might show a higher number of responders, but not a true additive effect and the drawn conclusion is not supported. 

We thank the reviewer for this important point. We have included Figure 2D and E as additional figures to show responders only and to make clear that there is a synergistic effect when only the responders are considered. The synergistic effect is seen more clearly in the responders of the mutated patients Figure 2 E. Please see the new figures 2D and E.

The figure 3 legend has been clarified.

We have added the following two articles which show, that Lenalidomide can enhance the immune effect of anti-PD-1 on line 120:

  1. Kwon et al. Two-Round Mixed Lymphocyte Reaction for Evaluation of the Functional Activities of Anti-PD-1 and Immunomodulators. Immune Netw. 2018 Dec 19;18(6):e45.  doi: 10.4110/in.2018.18.e45. eCollection 2018 Dec.
  2. Görgün et al. Lenalidomide Enhances Immune Checkpoint Blockade-Induced Immune Response in Multiple Myeloma. Clin Cancer Res. 2015 Oct 15;21(20):4607-18. doi: 10.1158/1078-0432.CCR-15-0200. Epub 2015 May 15.

Notably,  Figure 2 is not aligned, the legends in the manuscript look somewhat different and the tables included are not intuitive: what numbers are presented here? The percentage of CSF reduction for each individual patient? Also, Figure 3, first row seems to be missing some parts and the y axis is not readable for patients 2, 4, 10 and 11.

We appreciate this comment of the reviewer. We have adjusted the figure and clarified in the figure legend what is shown in the tables beneath Figure 2 A-C. We have also adjusted the axis of Figure 3.

  1. In context with the first point, the authors should discuss in more depth, how they expect these results to translate to clinical trials, as there have been basically two larger negative trials including Len in the induction treatment of AML patients (with of which have been quoted). Again, context and clarification are essential, to see novelty in this work. 

We thank the reviewer. To make it clearer in the text why we wanted to investigate whether Lenalidomide could work synergistically with anti-PD-1 to enhance anti-leukaemia responses in the context of leukaemia associated antigen stimulation, we have moved the pertinent part of the discussion to the Introduction (lines 118-126). This is highlighted in blue as it is not new text but does answer the reviewers comment.

  1. The ELISpot data needs some more context. The authors present data on three of sixteen tested patients. What happened with the other patients? How do the authors interpret these results?

We appreciate this comment of the reviewer. Due to a restricted number of patient cells, we could not include all patients into the ELISpot analyses. However, we have included a few more patients and also the cell-lines Oci-AML-2 (wildtype) and Oci-AML-3 (NPM1 mut) to make the picture clearer. In some cases, the ELISpot result correlated very well with the results from the CFI and in other cases not. We focus more on the Stem cell potential of our patient cells and do not want to overemphasize the ELISpot results. Please see new Figures 4 A and B and the new figure legends/ text passages.

Minor points

specific which anti-PD1 antibody was used

We appreciate the reviewer’s comment. We have specified this in some parts of the text, please see highlighted passages.

line 124: is this the same as Ficoll?

Yes exactly, we changed the text accordingly.

line 179: delete one  "colony forming".

We thank the reviewer and have changed the text accordingly.

lines 192: Sentence starting with "However": I do not understand this sentence; where is this shown?

We thank the reviewer, we have deleted this sentence.

line 255: stratify

We thank the reviewer, this has been corrected.

line 294: There are hardly any patients with a normal karyotype if analyzed with modern techniques (WGS, OGM); in my very personal opinion I would not emphasize this.

We appreciate this comment and we have deleted part of this sentence.

Round 2

Reviewer 2 Report

Thanks to the authors for incorporating my suggestions.

Apart from two last comments which I would ask the authors two include in the manuscript, I have no further objections. 

1)

line 405: this synergistic effect was clear...

Please change this to something like: "a synergistic effect was more pronounced" or "more visible"

2)

Line 483: NPM1 is not a mutation; "Pathogenic variants in NPM1 are among the most common in AML" 

Author Response

Thank you to the reviewer for their comments. These changes have been made as requested below.

'Apart from two last comments which I would ask the authors two include in the manuscript, I have no further objections. 

1)

line 405: this synergistic effect was clear...

Please change this to something like: "a synergistic effect was more pronounced" or "more visible"

Changed to 'However, a synergistic effect was clearer in the mutated patients'

2)

Line 483: NPM1 is not a mutation; "Pathogenic variants in NPM1 are among the most common in AML" 

Changed as suggested on line 393. 
